# Assessment of mastitis in camel using high-throughput sequencing

Rita Rahmeh [1]*, Abrar Akbar[1], Husam Alomirah[1], Mohamed Kishk[1], Abdulaziz Al-Ateeqi[1], Anisha Shajan[1], Thnayan Alonaizi[1], Alfonso Esposito[2]

1 Environment & Life Sciences Research Center, Kuwait Institute for Scientific Research, Kuwait City, Kuwait, 2 International Centre for Genetic Engineering and Biotechnology, Trieste, Italy

* rrahmeh@kisr.edu.kw

**Data Availability Statement:** The high-throughput sequencing data have been deposited in the NCBI database under the Bioproject PRJNA814002.

## Abstract

Camel milk is recognized as a functional food with significant economic value. Mastitis is one of the most common and costly diseases in the dairy industry. Mastitis, which is caused by pathogens such as bacteria, viruses, fungi, and algae, has an impact on the quality and quantity of milk produced as well as animal health and welfare. There is a paucity of data on the etiological factors that cause camel mastitis. This study reports the bacterial and fungal community involved in clinical camel mastitis using Illumina amplicon sequencing. A total of 25 milk samples were analyzed, including 9 samples with mastitis and 16 healthy samples. The bacterial community in healthy samples was significantly more diverse and abundant than in mastitis samples. The fungal population in mastitis samples, on the other hand, was more diverse and abundant. As compared to healthy samples, the genera *Staphylococcus*, *Streptococcus*, *Schlegelella*, *unclassified Enterobacteriaceae*, *Lactococcus*, *Jeotgalicoccus*. and *Klebsiella* were found to be abundant in mastitic milk. However, the genera *Corynebacterium*, *Enteractinococcus*, *unclassified Sphingomonadaceae*, *Atopostipes*, *Paenibacillus*, *Pseudomonas*, *Lactobacillus*, *Sphingomonas*, *Pediococcus* and *Moraxella* were reduced. In the fungal community, mastitis caused a significant increase in the relative abundance of the majority of taxa, including *Candida*, *Phanerochaete*, *Aspergillus*, *Cladosporium* and *unclassified Pyronemataceae*, while *Penicillium* and *Alternaria* showed a decline in relative abundance. In the bacterial and fungal communities, the discriminant analysis showed 19 and 5 differently abundant genera in healthy milk and mastitic milk, respectively. In conclusion, this study showed a microbiome dysbiosis linked to clinical camel mastitis, with opportunistic pathogens outgrowing commensal bacteria that were reduced. These findings are essential in designing an appropriate control program in the camel dairy herd, as well as in preventing and treating camel mastitis.

## Introduction

Dromedary camels are multipurpose animals having the ability to survive the harsh climate desert conditions. They represent about 95% of the camelid population [1]. Camels constitute an important source of milk in the world especially in Asian and African regions. The

**Funding:** This project was funded by Kuwait Foundation for the Advancement of Sciences (KFAS) under the project code PR18-12SL-16. This project was also funded by Kuwait Institute for Scientific Research (KISR) under the project code FB137C. RR received the funding for this work. The funders had no role in study design, data collection and analysis, decision to publish, or preparation of the manuscript.

**Competing interests:** The authors declare that they have no competing interests.

production of camel milk for human consumption was estimated to be 3.5 million tons. Camel milk is highly nutritional and deemed better than bovine milk. It is considered as functional food optimal for infant formula and for elderly. Beside its nutritional value, camel milk has therapeutic value against diabetes, asthma and jaundice [2, 3]. These traits increase the value of this milk and the local and international demand for it. As a result, camel dairy farms have been established worldwide with a steady integration into national and global economies [4, 5].

Mastitis is defined as the inflammation of the mammary gland and is considered the most critical disease in the dairy industry worldwide [6]. Mastitis can cause economic loss due to reducing milk production, deteriorated milk quality, lower probability of conception, higher treatment cost, and transmission of the disease to other species of animals [7]. Mastitis is usually epidemiologically categorized on the basis of source of infection into environmental mastitis caused by bacteria residing in the surrounding environment and contagious mastitis where the udders of the infected animal serve as a major reservoir of the pathogens which can spread from one animal to another during the milking process [8, 9].

Camels, like other dairy animals, can be affected by mastitis. This disease causes suffering for camels and poses a public health risk. According to the symptoms, mastitis in camels can also be classified as clinical or subclinical mastitis. Clinical mastitis is characterized by redness, heat and swelling of the mammary gland. However, subclinical mastitis is characterized by a lack of visible typical mastitis symptoms signs in the milk or in the udder [10, 11]. The major problems overwhelming the status of udder health in camels are tick infestation of the udder and thorny bushes, unhygienic practices, and cauterizations of the udder skin [12]. The occurrence of mastitis among lactating camels is being reported in some countries, including Somalia [13], Sudan [14], Kenya [15], and different parts of Ethiopia [16–18]. According to various studies, 45.66% of the world's camel population suffers from mastitis, with an average prevalence of 43% in Saudi Arabia and 24% in the United Arab Emirates [19]. However, the prevalence of camel mastitis in Kuwait is not reported despite the existence of contributing risk factors of mastitis that have a strong correlation with mastitis occurrence. These factors include nomadic housing of camel and improper hygienic measures during the milking process [20].

Mastitis is a complex disease caused by infections of a variety of microorganisms. Bacterial infections are considered the primary cause of mastitis in domestic animals. However, mycotic mastitis has long been known to be caused by fungi such as *Candida* genus infecting the mammary gland. [21]. Mastitis-causing pathogens in dairy animals and humans were determined in several studies using culture-independent approaches. The major causes of bovine mastitis have been well-studied [6]. Furthermore, human milk microbial community in lactational mastitis has been reported [22]. However, little is known about pathogens involved in mastitis occurrence in camels using culture-independent approaches. The available information on clinical and subclinical mastitis-causing microorganisms was exclusively reported based on conventional bacteriological (culture-dependent) methods, mainly from Ethiopia [11, 16, 23–25]. *Staphylococcus* and *Streptococcus* species were shown to be the most common causes of mastitis in Ethiopia [16, 23] while *Streptococcus agalactiae* was the only bacterium isolated from gangrenous mastitis in dromedary camels in the United Arab Emirates [11]. Furthermore, a high prevalence of fungi was reported in camel mastitis in Saudi Arabia representing 10% of the microbial isolates including *Candida*, *Cryptococcus*, *Trichosporon* and *Aspergillus* [26]. Mycotic mastitis caused by *Candida albicans* has been reported in a she-camel in Ethiopia [27].

Currently, it is well-known that milk harbors a wide range of bacteria including many which cannot be identified by culturing, leaving those microorganisms undetectable [28]. Thus, metagenomics approaches have been applied to investigate the microbiome of healthy

milk samples, as well as clinical and subclinical mastitis in human, bovine, and buffalo [22, 28, 29]. The application of a next-generation sequencing approach for profiling the bacterial community responsible for camel mastitis has an important impact on the clinical resolution of this disease and is essential for formulating strategies for controlling and preventing the development of this disease. Thus, the aim of this study was to identify the bacterial and fungal population involved in camel mastitis using Illumina amplicon sequencing.

## Materials and methods

All experimental protocols were approved by the Center Proposal Evaluation Committee (PEC) of Kuwait Institute for scientific research. The project was approved by Kuwait Foundation for the Advancement of Sciences (KFAS) under the project code PR18-12SL-16. All methods were performed in accordance with relevant institutional guidelines and regulations in compliance with the standards of animal rights and with camel owners' permission. Information about camels age, parity, and management are presented in S1 Table in S1 Appendix.

### Sample collection

Camel milk samples were collected by manual milking of individual dromedary camels belonging to camel owners in Kuwait desert. After visual and manual examination of the camels by a certified veterinarian, based on local signs and symptoms of mastitis, sixteen samples were collected from healthy animals with no clinical signs of mastitis. Nine samples were collected from camels showing signs of mastitis (abnormal udder such as redness, heat, swelling and reduced milk yield with blood in some samples). After owner permission was given, disinfection with 70% ethylic alcohol of the udder and the teats was performed by physical scrubbing and the first drops of milk were discarded. The samples were collected into sterile containers, labeled and delivered in an icebox to the laboratory for Somatic Cell Count (SCC) and DNA extraction. The SCC for all milk samples was performed immediately upon arrival at the laboratory using a direct cell counter (DCC, DeLaval International AB). The optimum SCC threshold for healthy milk is estimated to be around 200,000 cells/ml [30, 31]. The milk samples with SCC less than 200,000 cells/ml were considered healthy milk samples.

### DNA extraction

DNA extraction was performed at the same day of collection using GenElute Bacterial Genomic DNA Kit (Sigma-Aldrich) following the manufacturer's recommendations. Before extraction, 2 ml of milk was mixed and centrifuged at $10,000 \times g$ for 10 min. Then, genomic DNA was extracted from the pellet. The quantity of the extracted DNA was measured using a Qubit 3.0 Fluorometer (Invitrogen).

### Amplification of the 16S V3–V4 and ITS2 regions and sequencing

Custom fusion primers were used to prepare the 16S V3–V4 and ITS2 libraries. The custom primers included the appropriate P5/P7 Illumina adapter sequence, an 8-nt index sequence, and the gene-specific primer sequence for bacteria 341F (`ACTCCTACGGGAGGCAGCAG`) and 806R (`GGACTACHVGGGTWTCTAAT`) or the gene-specific primer sequence for fungi ITS3 (`GCATCGATGAAGAACGCAGC`) and ITS4 (`TCCTCCGCTTATTGATATGC`). Agencourt AMPure XP beads (Beckman Coulter) were used to purify the libraries, which were validated with an Agilent Technologies 2100 bioanalyzer. Sequencing was carried out on Illumina HiSeq 2500 in $2 \times 300$ bp mode. The high-throughput sequencing data are available at the NCBI database under the Bioproject PRJNA814002.

## Bioinformatics processing

Raw sequencing data were demultiplexed and amplicon sequence variants (ASVs) were determined with the DADA2 pipeline [32] implemented in R package dada2 v1.20.0. The BBTools package version 38.45 was used to identify and eliminate primer sequences from the 5' and 3' ends of input read pairs [33]. Forward and reverse reads with ≤ 2 and 4 expected errors, respectively, were retained. Contiguous sequences (contigs) were created by joining error-corrected reads with a minimum overlap of 20 base pairs. The "consensus" procedure used in DADA2 was employed to eliminate chimeric contigs that were consisted of two partial sequences of different origins. The IDTAXA approach [34] implemented in the R package DECIPHER v2.18.1 [35] was used to taxonomically classify the remaining contigs (ASVs) using the SILVA database SSU release 138 [36, 37]. ASVs with a classification confidence value ≥ 51% were retained. The R package DECIPHER was used to construct a neighbor-joining phylogenetic tree from aligned ASVs [38]. ASV copy numbers were inferred by hidden-state prediction in order to provide a better estimate of true ASV abundances, [39, 40]. Copy numbers were set to 1 for ASVs with a Nearest Sequenced Taxon Index (NSTI) > 2. ASV counts were normalized by dividing them by their respective copy numbers. To maintain the total count per sample, the result was multiplied by a sample-specific factor. Then, singletons were maintained while the counts were rounded to make them integers.

## Statistical analysis

The effect of treatment on individual ASVs was tested for statistical significance by R package metagenome Seq v1.31.0 (Paulson et al., 2019) using a zero-inflated Gaussian mixture model (Paulson et al., 2013). ASVs with at least ten counts in two or more samples were considered. The effect size was expressed as log2(fold-change) of size-factor-normalized ASV counts. Log Odds Differential Abundance (log(ODA)) gives the log-odds that the abundance of the ASV differs according to the considered condition. P-values for tests within the same comparison level were adjusted using Benjamini-Hochberg correction (false discovery rate) [41].

The indices calculated with R package vegan v2.6.0 were used to measure alpha diversity as follows: Observed ASVs; Chao1 richness; ACE (Abundance-based Coverage Estimator) richness, Shannon diversity; Simpson diversity; Inverse Simpson diversity; and Effective richness $^1$D [42]. Descriptive variables were used for sample grouping. Rarefaction curves were produced based on observed ASVs.

Beta diversity was evaluated. R package phyloSeq v1.37.0 was used to determine the extent of change in ASV abundances across samples. A Principal Component Analysis (PCoA) was performed on both unweighted and weighted UniFrac dissimilarities [43] derived from ASV counts and phylogenetic information. R package DESeq2 v1.30.1 was used to determine the extent of change in ASV abundances across samples explained by grouping variables [44]. ASV counts were normalized by library size and transformed using the variance-stabilizing transformation (VST) [45]. Redundancy Analysis implemented in R package vegan v2.6.0 was performed on VST-transformed normalized ASV counts (ASV$_{trans}$) [42].

Using R package microbiome Marker v0.0.1.9000 [46], ASVs counts aggregated at different taxonomic ranks were subjected to linear discriminant analysis Effect Size (LEfSe) [47]. ASVs with at least ten counts in two or more samples were considered.

## Results

### Sequencing summary

This study included 25 milk samples collected from dromedary camels in Kuwait desert, with 16 healthy with SCC less than 200,000 cells/ml and 9 mastitis samples. The samples were

sequenced in the V3-V4 region of 16S rRNA gene, resulting in 2.35 million total read pairs. A total of 1.75 million high-quality 16S rRNA gene sequences (ASVs) were retained for the samples. Of the 1.75 million ASVs with taxonomy, 63.27% were attributed to the healthy samples and 36.73% for mastitis samples. Samples exhibited a median of 167 ASVs (inter-quartile range (IQR) = 107–243). The per-sample numbers of read pairs remaining after each pipeline step are summarized in S2 Table in S1 Appendix.

The sequencing of the ITS region of eighteen samples generated 2.31 million total read pairs. A total of 1.97 million high-quality sequences were retained for the samples. Samples exhibited a median of 32 ASVs (IQR = 22–59). Of the 1.97 million ASVs with taxonomy, 80.9% were attributed to the healthy samples and 19.1% for mastitis samples. The per-sample numbers of read pairs remaining after each pipeline step are summarized in S3 Table in S1 Appendix.

## Camel milk microbiota in healthy and mastitis milk samples

The bacterial community in healthy camel milk samples were classified into 26 phyla while mastitis samples comprised 16 phyla. The healthy camel milk microbiota was dominated by the phyla *Proteobacteria* and *Firmicutes* accounting for 38.14% and 34.06% of the bacterial population, respectively, followed by *Actinobacteria* (20.54%) and *Bacteroidota* (5.71%) (Table 1). Although *Fusobacteriota*, *Patescibacteria* and *Deinococcota* were minor phyla, they accounted for more than 0.1% of the overall bacterial population. Mastitis milk samples were dominated by the phyla *proteobacteria* (37.97%) and *Firmicutes* (40.29%). As compared to healthy milk samples, mastitis samples showed an increase in *Firmicutes* (40.29% vs. 34.06%) and a decrease in *Actinobacteria* (17.46% vs. 20.54%) and *Bacteroidota* (3.69% vs. 5.71%). Uncultured bacteria (bacteria that cannot be cultured using standard methods) accounted for 0.2% and 0.01% of the total bacteria in healthy and mastitis milk samples, respectively. Rare taxa, which were defined as having a mean abundance of <1% across all samples, were summarized as "others" (Fig 1).

At the genus level, the microbiota of healthy camel milk was dominated by *Glutamicibacter* (6.98%) and *Schlegelella* (6.74%), followed by the genera *Acinetobacter* (6.48%), *Pseudomonas* (5.9%), *Lactobacillus* (5.29%), *Staphylococcus* (5.27%), *Paenibacillus* (4.53%), *Corynebacterium* (4.02%) and *Streptococcus* (3.21%) (Table 2). The microbiota of mastitis camel milk was dominated by the genera *Streptococcus* (12.21%) and *Schlegelella* (11.79%) followed by the genera *Staphylococcus* (9.97%), *uncl. Enterobacteriaceae* (8.18%), *Glutamicibacter* (7.75%), *Acinetobacter* (6.06%), *Pseudomonas* (3.51%) and *Paenibacillus* (3.83%). As compared to healthy milk samples, mastitis milk microbiota was enriched in *Staphylococcus* (9.97% vs. 5.27%), *Streptococcus* (12.21% vs. 3.21%), *Schlegelella* (11.79% vs. 6.74%), *unclassified Enterobacteriaceae* (8.18% vs. 0.75%), *Lactococcus* (3.24% vs. 0.01), and *Jeotgalicoccus* (2.33% vs. 1.48%). Mastitis -was also associated with the appearance of *Klebsiella* genus that was not detected in the healthy

**Table 1. Relative abundance of bacterial population in healthy and mastitis camel milk at phylum level for 16S rRNA.**

| Phylum | Mean Relative abundance 16S | |
| --- | --- | --- |
| | Healthy | Mastitis |
| *Proteobacteria* | 38.14 | 37.98 |
| *Firmicutes* | 34.07 | 40.29 |
| *Actinobacteriota* | 20.54 | 17.46 |
| *Bacteroidota* | 5.71 | 3.69 |
| *Others* | 1.54 | 0.58 |

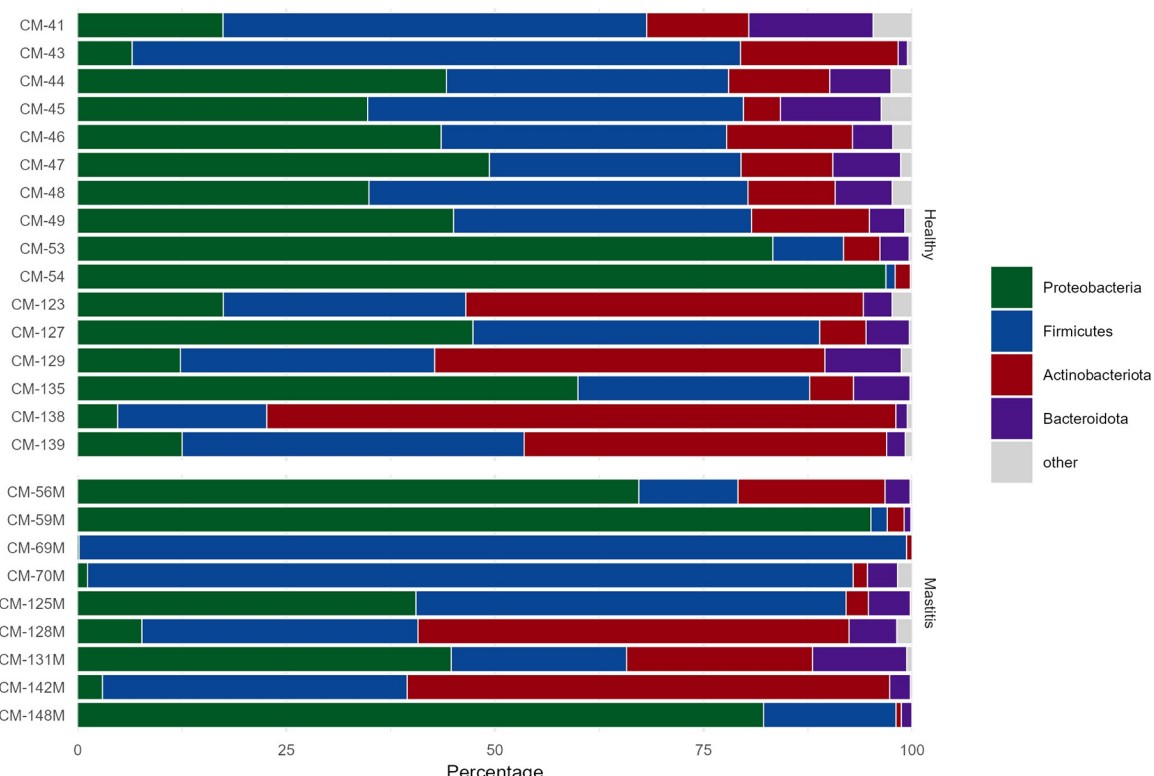

**Fig 1. Taxon frequencies (percent of total sample counts) of the most dominant bacterial population in healthy and mastitis camel milk summarized at the phylum level for the 16S rRNA.** Rare taxa with a mean abundance of <1% across all samples are summarized as "Others". M corresponds to the mastitis samples.

milk samples. Other genera, such as *Brachybacterium* and *Glutamicibacter* also increased slightly with mastitis. However, the genera *Corynebacterium*, *Enteractinococcus*, unclassified *Sphingomonadaceae*, *Atopostipes*, *Paenibacillus* and *Pseudomonas* decreased in the mastitis milk samples. As compared to healthy milk samples, some genera were present in low abundance in the mastitis samples; these genera included *Lactobacillus* (0.24% vs. 5.29%) *Sphingomonas* (0.16% vs. 2.16%), *Pediococcus* and *Moraxella* (0.4% vs. 2.6%) (Fig 2). Taxon frequencies of the most dominant bacterial population in healthy and mastitis camel milk samples at the phylum and genus level are presented in S4 and S5 Tables in S1 Appendix, respectively. Statistical differences of the effect of mastitis on individual ASVs are presented in S6 Table in S1 Appendix.

LefSe was next used to explain differences between classes in healthy and mastitis groups. LEfSe analysis identified 19 differentially abundant genera in healthy and mastitis. The genera *Lactobacillus*, *Moraxella*, *Pediococcus*, *Sphingomonas*, *Cutibacterium*, *Novosphingobium*, *Cnuella*, unclassified *Burkholderiales*, *Methylobacterium-Methylorubrum*, *Tepidimonas*, *Bacillus*, *Enterococcus*, unclassified *Streptococcaceae*, *Bergeyella*, *Lysobacter*, *TM7a*, *Dolosigranulum*, and *Clostridium sensu stricto 9* were significantly more enriched in the healthy group (P< 0.05). Alternatively, the genus *Exiguobacterium* was significantly more enriched in mastitis (Table 3). The effect size and statistical significance of the top ten marker taxa per sample group between healthy and mastitis groups is shown in Fig 3 and a Cladogram that represents differentially abundant microbiota at the class, order, family, and genus level between healthy and mastitis groups is shown in Fig 4.

The fungal community in healthy and mastitis camel milk samples was classified into 3 phyla, *Ascomycota*, *Basidiomycota* and *Mucoromycota* (Fig 5). As compared to healthy milk

**Table 2. Relative abundance of bacterial population in healthy and mastitis camel milk at genus level for 16S rRNA.**

| | Mean Relative abundance 16S (%) | |
|---|---|---|
| Genus | Healthy | Mastitis |
| *Schlegelella* | 6.74 | 11.79 |
| *Streptococcus* | 3.21 | 12.21 |
| *Staphylococcus* | 5.27 | 9.97 |
| *Glutamicibacter* | 6.98 | 7.75 |
| *Acinetobacter* | 6.48 | 6.06 |
| *Pseudomonas* | 5.90 | 3.51 |
| unclassified *Enterobacteriaceae* | 0.75 | 8.18 |
| *Paenibacillus* | 4.53 | 3.83 |
| *Corynebacterium* | 4.02 | 1.85 |
| *Lactobacillus* | 5.29 | 0.24 |
| *Jeotgalicoccus* | 1.48 | 2.33 |
| unclassified *Sphingomonadaceae* | 2.60 | 0.85 |
| *Brachybacterium* | 1.41 | 1.91 |
| *Lactococcus* | 0.01 | 3.24 |
| *Pediococcus* | 2.61 | 0.44 |
| unclassified *Comamonadaceae* | 1.81 | 1.19 |
| *Moraxella* | 2.57 | 0.42 |
| *Klebsiella* | 0.00 | 2.84 |
| *Enteractinococcus* | 1.56 | 0.98 |
| *Sphingomonas* | 2.16 | 0.16 |
| *Atopostipes* | 1.52 | 0.72 |
| *Others* | 33.10 | 19.51 |

samples, mastitis milk microbiota was enriched in *Basidiomycota* (16.48% vs. 2.35%) while a decrease in the relative abundance of *Ascomycota* (83.44% vs. 97.3%) was observed (Table 4). At the genus level, the fungal population of healthy camel milk was dominated by the genus *Penicillium* (76.93%), followed by *Cladosporium* (4.05%) and *Candida* (3.89%) (Table 5). The fungal population of mastitis camel milk was dominated by the genera *Penicillium* (36.82%), followed by *Candida* (13.6%), *Phanerochaete* (13.32%), *Cladosporium* (5.15%), and *Aspergillus* (5.33%). In mastitic milk, a remarkable increase in the relative abundance of the majority of the genera including *Candida*, *Phanerochaete*, *Aspergillus*, *Cladosporium* and unclassified *Pyronemataceae* was shown. However, a decrease in the relative abundance of *Penicillium* and *Alternaria* was observed (Fig 6).

LEfSe analysis identified one differentially abundant genera in healthy samples while four were identified in mastitis. The genera *Acremonium*, *Chaetomium*, *Wallemia*, and unclassified *Pleosporaceae* were significantly more enriched in the mastitis group (P< 0.05). Alternatively, the genus *Tylospora* was significantly more enriched in the healthy group Table 6. The effect size and statistical significance of the top ten marker taxa per sample group between healthy and mastitis groups is shown in Fig 7 and a Cladogram that represents differentially abundant microbiota at the class, order, family, and genus level between healthy and mastitis groups is shown in Fig 8.

## Microbial richness and diversity in healthy and mastitis milk samples

The median values of the alpha diversity indices for healthy and mastitis samples are presented in Table 7. For the bacterial population, the median values of the six indices were higher in

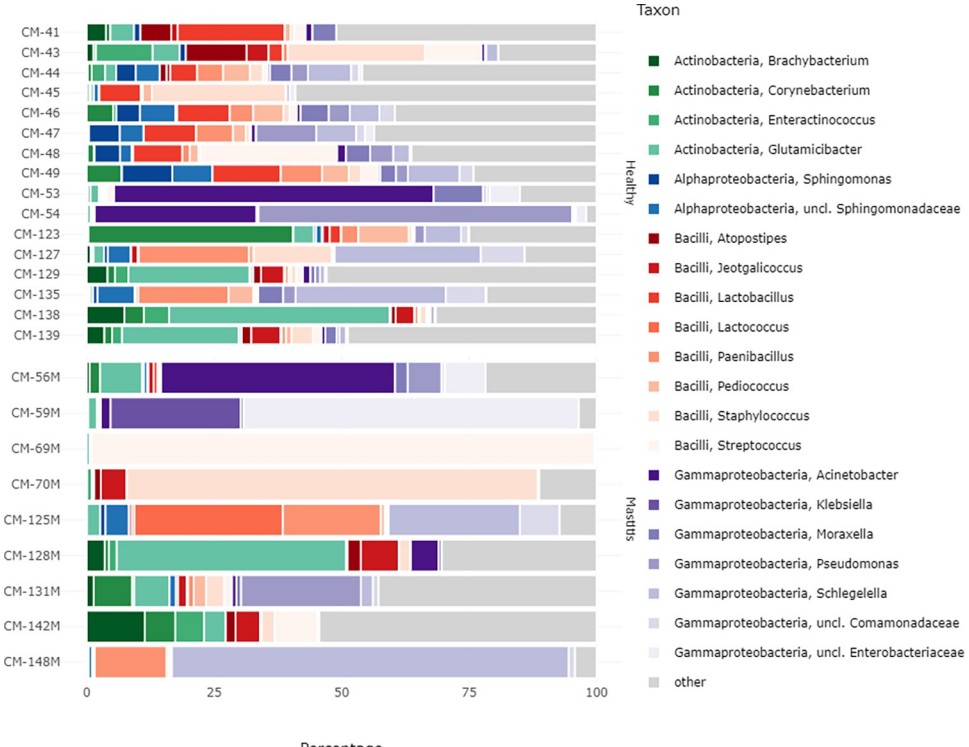

**Fig 2. Taxon frequencies (percentage of total sample counts) of the most dominant bacterial population in healthy and mastitis camel milk summarized at the genus level for the 16S rRNA.** Rare taxa with a mean abundance of <1% across all samples are summarized as "Others". M corresponds to the mastitis samples.

healthy samples than in mastitis samples. Shannon and Simpson diversity indices were higher in healthy samples, indicating that the healthy samples had higher bacterial diversity compared to mastitis samples. The bacterial richness was also compared by sample group, as estimated by the Chao1 richness index. Compared to mastitis milk samples, the healthy milk samples showed greater bacterial richness. The distribution of alpha diversity measures in relation to sample groups are shown in Fig 9. However, for the fungal population, the median values of the alpha diversity indices were higher in mastitis samples indicating a high diversity and richness in mastitis samples (Fig 10). Bacterial and fungal richness were also evaluated by rarefaction curves in sample groups. For healthy and mastitis groups, the rarefaction curves for all samples were saturated and reached a plateau, suggesting that the sequencing depth was enough to capture the majority of the bacteria and fungus present in healthy and mastitis camel milk samples. Furthermore, the bacterial richness in healthy milk samples was higher compared to mastitis milk samples (Fig 11A), in contrast to the fungal population (Fig 11B).

The beta diversity analysis was then conducted to compare healthy and mastitis milk samples. The extent of change in ASVs abundances across samples was determined using unweighted and weighted UniFrac distance metrics. For the bacterial population, PCoA for unweighted UniFrac bundled 30.9% of the total variance in the first three principal coordinates. The first principal coordinate (axis 1) explained the 15.2% of the variability and the second principal coordinate (axis 2) explained the 9.2% (Fig 12A). For weighted UniFrac, PCoA bundled 63.1% of the total variance in the first three principal coordinates. Axis 1 explained the 30.7% of the variability and axis 2 explained the 15.9% (Fig 12B). For unweighted and weighted UniFrac, axis 1 and 2 separated healthy from mastitis milk samples, although with some overlaps. For the fungal population, PCoA for unweighted UniFrac bundled 47.8% of

**Table 3. Lefse marker table at taxonomic level for healthy and mastitis samples for 16S rRNA.**

| Feature | Enriched Group | log10(LDA Score) | Adjusted P-value |
|---|---|---|---|
| f__Lactobacillaceae | Healthy | 4.9 | 0.0039 |
| c__Alphaproteobacteria | Healthy | 4.8 | 0.017 |
| g__Lactobacillus | Healthy | 4.8 | 0.027 |
| o__Sphingomonadales | Healthy | 4.7 | 0.036 |
| f__Sphingomonadaceae | Healthy | 4.7 | 0.036 |
| o__Paenibacillales | Healthy | 4.4 | 0.037 |
| f__Paenibacillaceae | Healthy | 4.4 | 0.037 |
| g__Moraxella | Healthy | 4.4 | 0.012 |
| g__Pediococcus | Healthy | 4.4 | 0.0068 |
| g__Sphingomonas | Healthy | 4.4 | 0.008 |
| o__Propionibacteriales | Healthy | 4.1 | 0.0063 |
| f__Propionibacteriaceae | Healthy | 4 | 0.015 |
| g__Cutibacterium | Healthy | 4 | 0.0033 |
| o__Rhizobiales | Healthy | 3.8 | 0.023 |
| g__Novosphingobium | Healthy | 3.8 | 0.018 |
| g__Cnuella | Healthy | 3.8 | 0.014 |
| f__uncl. Burkholderiales | Healthy | 3.7 | 0.024 |
| g__uncl. Burkholderiales | Healthy | 3.7 | 0.024 |
| f__Beijerinckiaceae | Healthy | 3.7 | 0.015 |
| g__Methylobacterium-Methylorubrum | Healthy | 3.7 | 0.02 |
| g__Tepidimonas | Healthy | 3.6 | 0.024 |
| f__Oxalobacteraceae | Healthy | 3.3 | 0.025 |
| g__Bacillus | Healthy | 3.2 | 0.0076 |
| g__Enterococcus | Healthy | 3.2 | 0.0014 |
| f__Enterococcaceae | Healthy | 3.2 | 0.0014 |
| o__Saccharimonadales | Healthy | 3.2 | 0.044 |
| f__Saccharimonadaceae | Healthy | 3.2 | 0.044 |
| c__Saccharimonadia | Healthy | 3.2 | 0.044 |
| g__uncl. Streptococcaceae | Healthy | 3.1 | 0.038 |
| g__Bergeyella | Healthy | 3 | 0.041 |
| g__Lysobacter | Healthy | 3 | 0.035 |
| g__TM7a | Healthy | 3 | 0.041 |
| o__Frankiales | Healthy | 2.9 | 0.014 |
| f__Geodermatophilaceae | Healthy | 2.8 | 0.041 |
| g__Dolosigranulum | Healthy | 2.6 | 0.041 |
| g__Clostridium sensu stricto 9 | Healthy | 2.4 | 0.041 |
| f__Exiguobacteraceae | Mastitis | 3.2 | 0.016 |
| g__Exiguobacterium | Mastitis | 3.2 | 0.016 |
| o__Exiguobacterales | Mastitis | 3.2 | 0.016 |

the total variance in the first three principal coordinates (Fig 12C). For weighted UniFrac, PCoA bundled 86.1% of the total variance in the first three principal coordinates (Fig 12D).

To study the statistical effect of mastitis on the milk microbial community, a Redundancy Analysis (RDA) was performed on VST-transformed normalized ASV counts. The resulting RDA model explained 2.7% of the total variance in ASVtrans. Given the study's small sample size, a 90% confidence interval was considered; milk bacterial community was significantly different in mastitis (RDA test; P = 0.056).

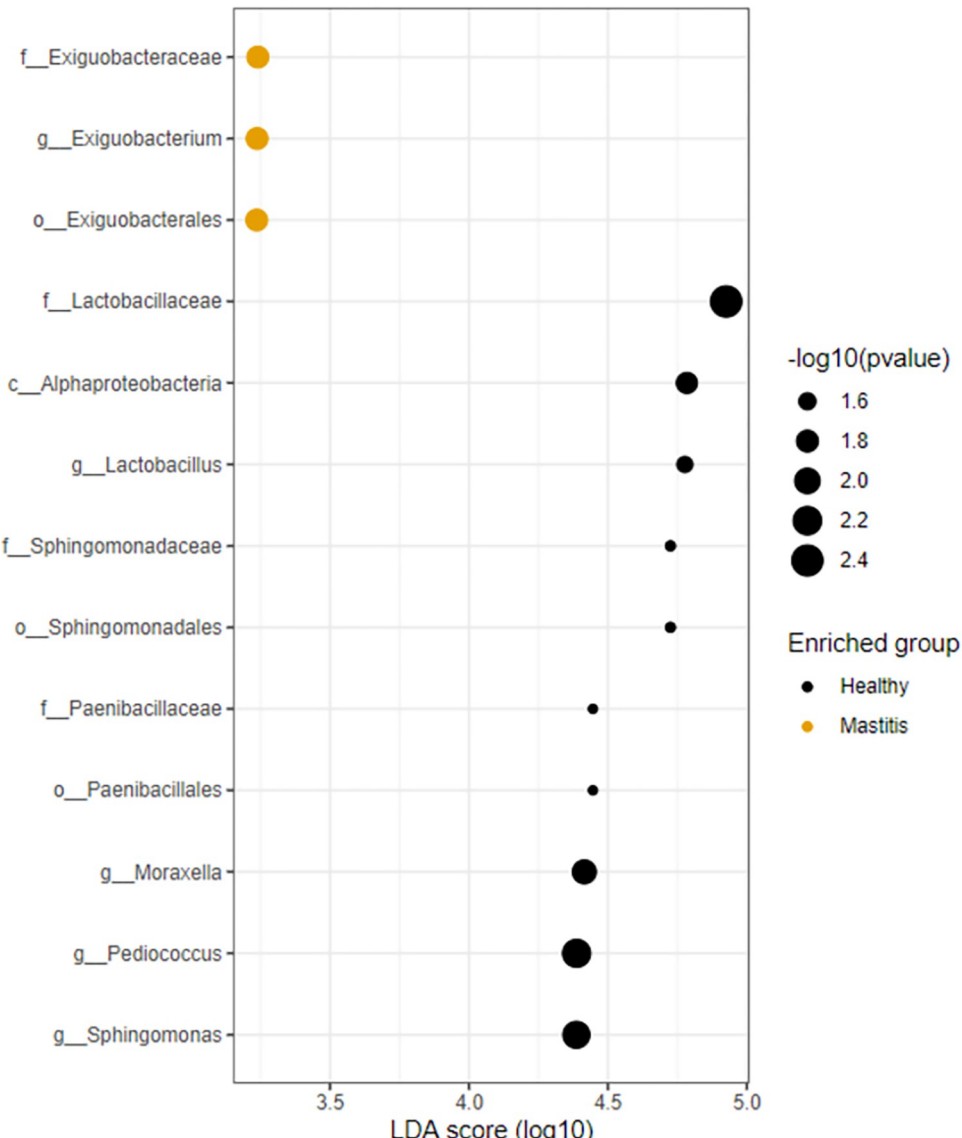

**Fig 3. Effect size and statistical significance of the top ten marker per sample group corresponding to healthy and mastitis samples for the 16S rRNA.**

## Discussion

Camel milk has nutritional value and medicinal properties against a range of diseases; hence it is gaining international recognition. Camel mastitis is a complex disease that affects the amount and quality of milk produced. Bacterial and/or fungal infections are among the etiological agents of camel mastitis. Metagenomics approaches have been applied to investigate the microbiome of healthy milk samples, as well as clinical and subclinical mastitis in human, bovine, and buffalo [22, 28, 29]. However, knowledge regarding the pathogens responsible for camel milk mastitis is limited and exclusively based on conventional bacteriological methods. Till date there are no studies reporting the microbial population associated with camel mastitis through culture-independent approaches. This study reports the first detailed characterization of milk microbiota in camel milk with mastitis as determined by 16S rRNA gene sequencing.

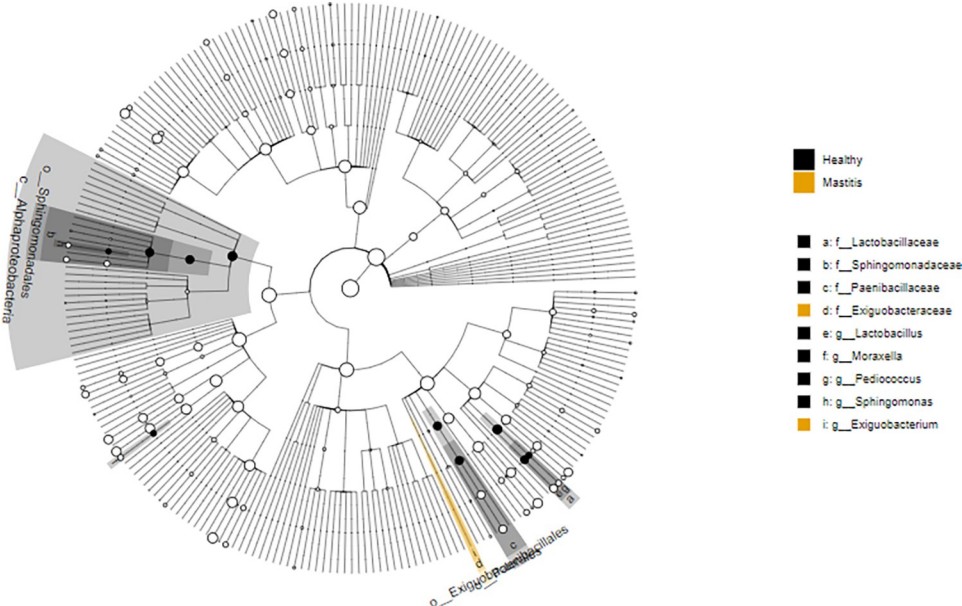

**Fig 4. Cladogram of the top ten marker taxa per sample group corresponding to healthy and mastitis samples for the 16S rRNA.**

Many pathogens live in large communities alongside other microbes, all of which participate in intricate interactions that might influence or drive disease processes [48–50]. Some commensal bacteria might become pathogenic under the impact of several conditions, such as the actions of other microorganisms or antibiotics that cause shifts in microbiome

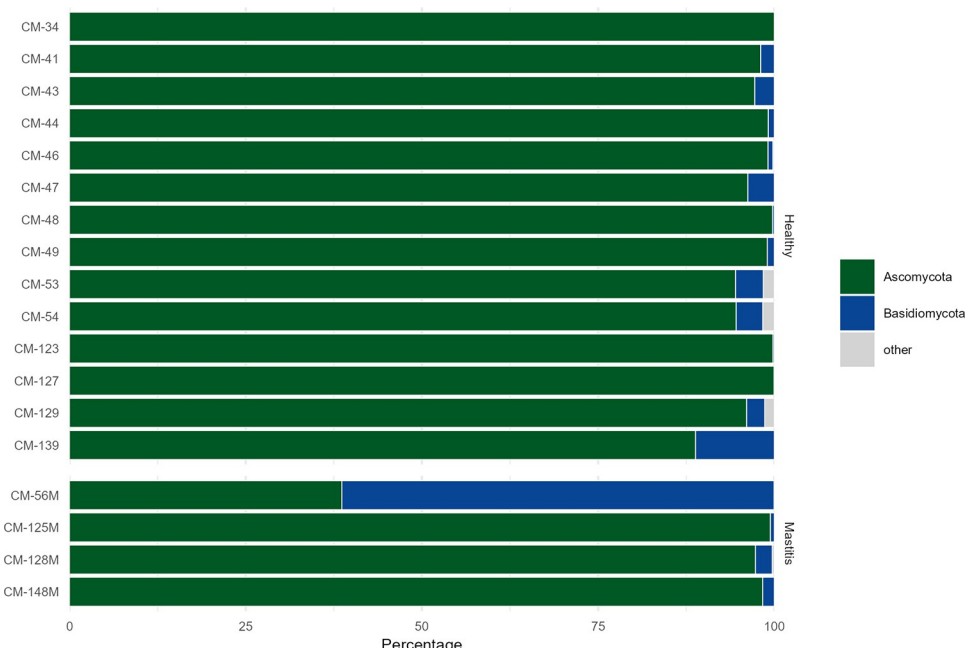

**Fig 5. Taxon frequencies (percentage of total sample counts) of the most dominant bacterial population in healthy and mastitis camel milk summarized at the phylum level for the ITS.** Rare taxa with a mean abundance of <1% across all samples are summarized as "Others". M corresponds to the mastitis milk samples.

**Table 4. Relative abundance of fungal population in healthy and mastitis camel milk at phylum level for ITS.**

| Phylum | Mean Relative abundance ITS | |
|---|---|---|
| | Healthy | Mastitis |
| *Ascomycota* | 97.31 | 83.44 |
| *Basidiomycota* | 2.35 | 16.48 |
| *uncl. Fungi* | 0.24 | 0.01 |
| *Mucoromycota* | 0.09 | 0.08 |

composition [51]. As exemplified by several metagenomics studies, the microbiome of healthy individuals includes pathogenic bacteria, and a relationship was found between resident microbes and the development of disease [28, 50]. As our results indicated, the genera *Staphylococcus* and *Streptococcus*, alongside other opportunistic bacteria were also found in healthy camel milk, suggesting that the development of mastitis in camels is mainly due to an imbalance or shift in the microbial population, rather than a primary infection. We observed an increase in the abundance of *Staphylococcus*, *Streptococcus*, *Schlegelella*, and *unclassified Enterobacteriaceae*, *Klebsiella*, and *Jeotgalicoccus* associated with mastitis. These results are consistent with previous culture-based studies on camel milk microbiota where *Staphylococcus* and *Streptococcus* were shown to be the most common causes of mastitis in Ethiopia [16, 23]. *Streptococcus agalactiae* was the only bacterium isolated from gangrenous mastitis in dromedary camels in the United Arab Emirates [11]. *Staphylococcus* and *Streptococcus* have been also known to be the main cause of human and bovine mastitis. *Staphylococcus aureus* has long been thought to be the main causative agent of mastitis; nevertheless, *Staphylococcus epidermidis* is becoming a leading cause of subacute and acute mastitis in both humans and animals [52]. In accordance to our study, an assessment of the microbial community in human milk showed that genera including *Staphylococcus* and *Klebsiella* were significantly enriched in mastitis samples. As opposed to mastitis human milk, the genus *Pseudomonas* decreased in camel mastitis with no change in the *Acinetobacter* relative abundance [22]. *Pseudomonas* genus present in healthy camel milk with a relative abundance (5.9%) was also found in the core microbiota of healthy bovine and buffalo milk, with a lower relative abundance than in bovine milk (18.75%) and a higher relative abundance than in buffalo milk (1.5%) [28, 53]. The genera *Jeotgalicoccus* was present in the majority of the mastitis samples, this genus along with *Schlegelella* were found for the first time in mastitis camel milk. The genus *Exiguobacterium* was identified as differentially abundant genera in mastitis samples by LefSe analysis. The enriched genus *Exiguobacterium* was found in a previous study among the bacteria identified from

**Table 5. Relative abundance of fungal population in healthy and mastitis camel milk at genus level for ITS.**

| Genus | Mean Relative abundance ITS | |
|---|---|---|
| | Healthy | Mastitis |
| *Penicillium* | 76.93 | 36.82 |
| *Candida* | 3.89 | 13.60 |
| *Phanerochaete* | 0.00 | 13.32 |
| *Cladosporium* | 4.05 | 5.15 |
| *Aspergillus* | 3.28 | 5.33 |
| *unclassified Pyronemataceae* | 0.81 | 2.38 |
| *Fusarium* | 1.21 | 1.65 |
| *Alternaria* | 1.62 | 0.76 |
| *Others* | 8.21 | 20.99 |

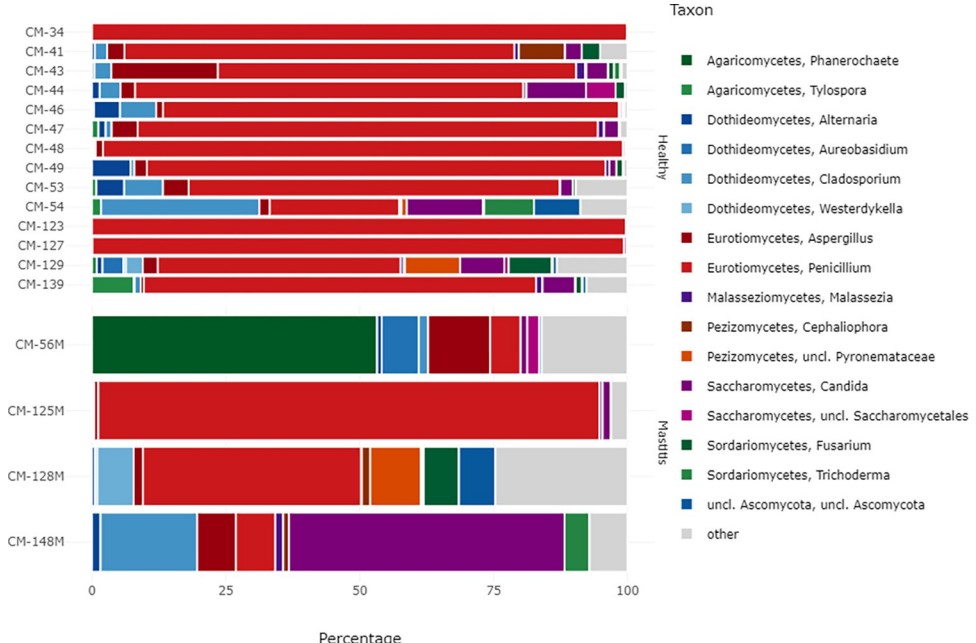

**Fig 6. Taxon frequencies (percentage of total sample counts) of the most dominant bacterial population in healthy and mastitis camel milk summarized at the genus level for the ITS.** Rare taxa with a mean abundance of <1% across all samples are summarized as "Others". M corresponds to the mastitis samples.

subclinical mastitis milk samples of Kankrej cows in India using pyrosequencing [54]. In addition, the presence of this halophilic bacteria was previously associated with goat and sheep milk [55, 56]. It is noteworthy that this study demonstrates a diversity in camel mastitis etiology. The increase in the abundance of *Staphylococcus* and *Streptococcus* is mainly associated to two samples CM-69M and CM-70M originating from camels suffering from severe clinical mastitis. To fully understand the mastitis-causing pathogens in camels, additional investigations are required, including a large number of mastitic milk samples along with bacterial culturing.

In parallel to an increase in the opportunistic bacteria, a significant decrease in the commensal beneficial bacteria such as lactic acid bacteria (*Lactobacillus*, *Pediococcus*, *Enterococcus*) was observed in mastitis. These lactic acid bacteria were identified as differentially abundant

**Table 6. Lefse marker table at taxonomic level for healthy and mastitis samples for ITS.**

| Feature | Enriched Group | log10(LDA Score) | Adjusted P-value |
|---|---|---|---|
| g__*Tylospora* | Healthy | 4.1 | 0.026 |
| o__*Atheliales* | Healthy | 4.1 | 0.026 |
| f__*Atheliaceae* | Healthy | 4.1 | 0.026 |
| g__*unclassified Pleosporaceae* | Mastitis | 4.2 | 0.034 |
| f__*Hypocreales fam Incertae sedis* | Mastitis | 3.6 | 0.022 |
| g__*Acremonium* | Mastitis | 3.6 | 0.034 |
| g__*Chaetomium* | Mastitis | 3.3 | 0.05 |
| g__*Wallemia* | Mastitis | 3.1 | 0.0065 |
| o__*Wallemiales* | Mastitis | 3.1 | 0.0065 |
| f__*Wallemiaceae* | Mastitis | 3 | 0.0065 |
| c__*Wallemiomycetes* | Mastitis | 3 | 0.0065 |

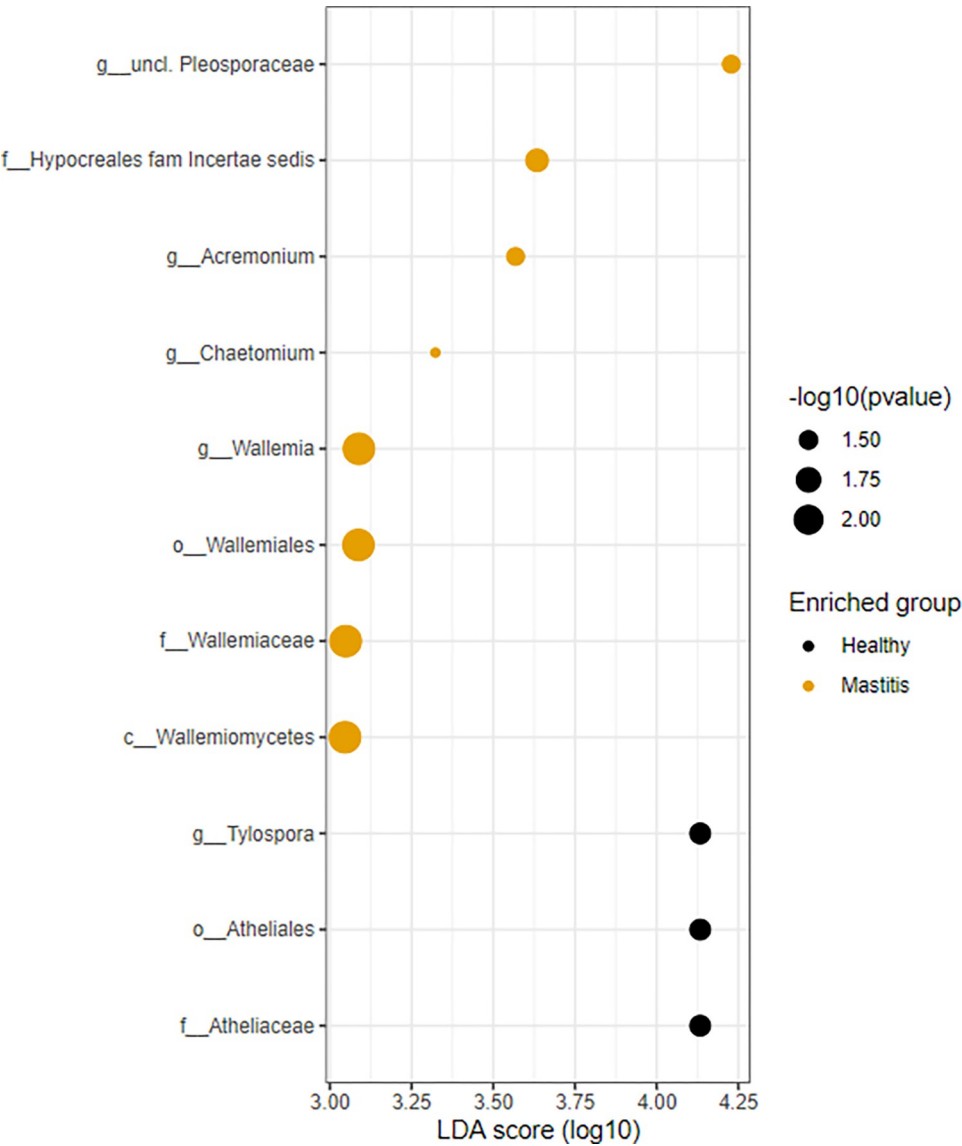

**Fig 7. Effect size and statistical significance of the top ten marker per sample group corresponding to healthy and mastitis samples for the ITS.**

genera in healthy samples by LefSe analysis. However, an increase in the *Lactococcus* genus was observed. Similarly, an increase in this genus was observed in water buffalo milk during mastitis [28]. Lactic acid bacteria have therapeutic properties, making them potential candidates for inclusion in a probiotic cocktail to prevent and/or treat infectious mastitis in camels [57]. The use of probiotics has piqued the curiosity of veterinarians. To treat cow mastitis, a bacteriocin-producing *Lactococcus lactis* was proven equally effective *in vivo* as a standard antibiotic treatment [58].

Fungal-related mastitis was reported previously in several studies. The incidence of this type of mastitis is usually low in dairy herds, but it has significantly increased during the last decade. Teat injuries may predispose to establishment of a fungal infection [21]. The fungal genera identified in this study, *Candida*, *Aspergillus* and *Cladosporium* have also been described in a previous study conducted on cows suffering from mastitis in Poland. Among

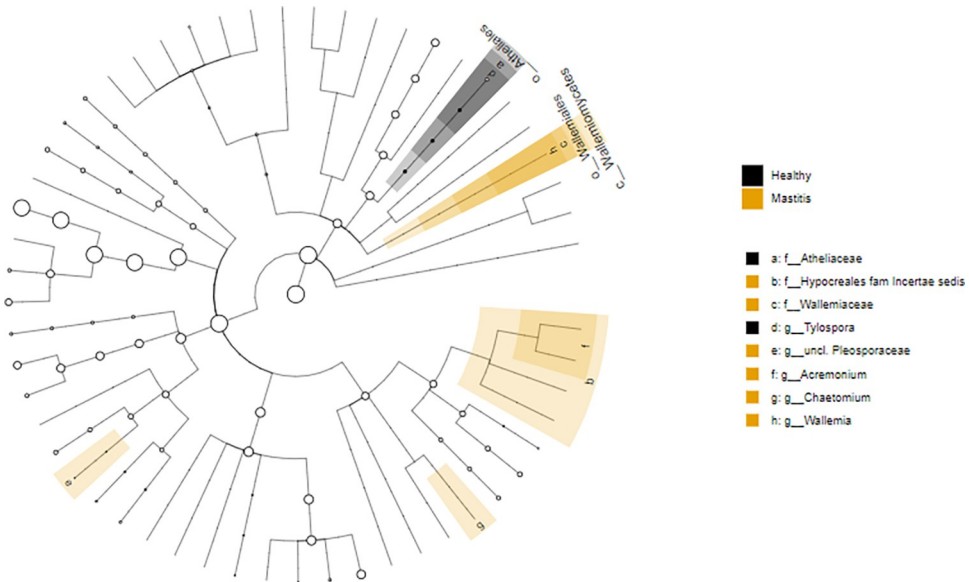

**Fig 8. Cladogram of the top ten marker taxa per sample group corresponding to healthy and mastitis samples for the ITS.**

fungi, *Candida* is the predominant cause of mycotic mastitis. Seven species of *Candida* were suggested as etiological agents of bovine mastitis in Poland [21]. *Candida* species played a pathogenic role in mycotic mastitis of cows in China [59]. Pal et al., 2015 suggested a role of *Candida* in mastitis of one humped camel in Ethiopia by applying standard mycological techniques [27]. The genus *Aspergillus* was the most predominant fungi followed by *Candida* in she camel mastitis in Egypt in a culture-dependent study [60]. These two genera were also isolated from goat subclinical and clinical mastitis [61–63]. *Cladosporium* was detected among the mycotic causes of subclinical bovine mastitis in Egypt, ovine and caprine mastitis in Nigeria, and goat mastitis in Iraq [61, 64, 65]. However, *Phanerochaete* genus was not previously reported in dairy animal mastitis.

The diversity and richness of the bacterial community in healthy milk samples was considerably greater than that of mastitis milk samples, according to rarefaction curves and alpha diversity indices. Our findings are consistent with previous studies that showed that the microbiome of bovine and buffalo milk in healthy samples is more diverse than that reported in mastitis [6, 28, 29]. However, we observed increased diversity and richness of the fungal population of milk from mastitis samples in comparison to healthy samples. According to the literature, in comparison to bacterial biodiversity, knowledge of fungal diversity in raw milk is limited. The beta diversity analysis showed that the percentage of variance in the first three principal coordinate for the weighted Unifrac was greater than unweighted UniFrac, suggesting that the diversity could have probably resulted from the abundance, rather than the presence/absence of ASVs.

**Table 7. Median alpha diversity measures by sample group for healthy and mastitis milk samples for 16S rRNA.**

| Treatment | Observed ASVs | Chao1 Richness | ACE Richness | Shannon Diversity | Simpson Diversity | Inverse Simpson Diversity | Effective Richness |
|---|---|---|---|---|---|---|---|
| Healthy (16S) | 204.5 | 204.5 | 204.5 | 3.71 | 0.94 | 16.62 | 41.08 |
| Mastitis (16S) | 107 | 107 | 108.58 | 2.27 | 0.78 | 4.5 | 9.64 |
| Healthy (ITS) | 29 | 29 | 33 | 1.04 | 0.37 | 1.62 | 2.91 |
| Mastitis (ITS) | 46 | 46 | 30 | 2.02 | 0.72 | 3.55 | 7.79 |

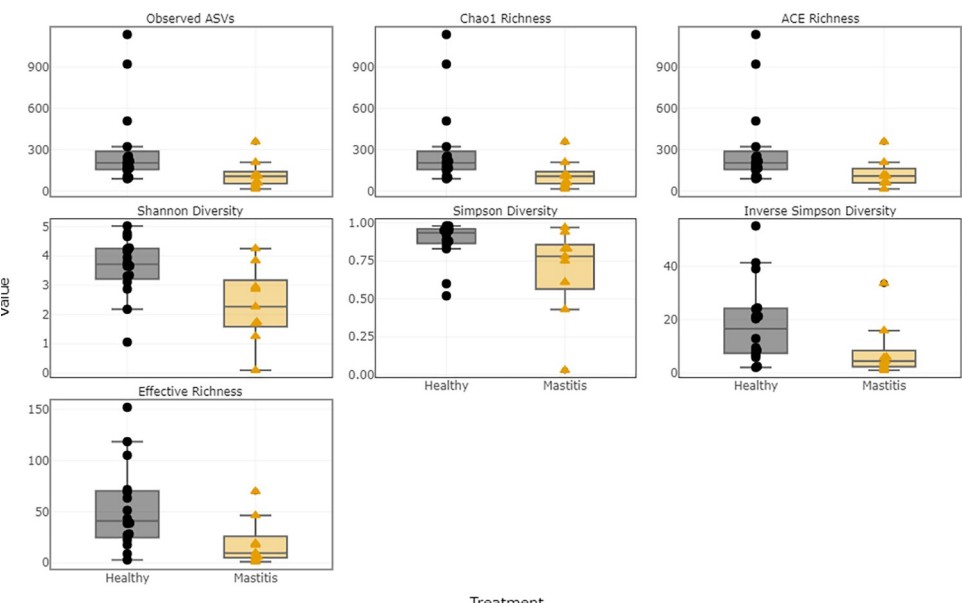

**Fig 9. Distribution of alpha diversity measures in relation to sample groups for the 16S rRNA.**

Regarding the potential origin of the microbiota detected in camel milk of healthy and mastitis groups, the majority of the detected genera are most probably originating from the dairy environment. An extensive study was conducted to link the bacteria discovered in cow milk to their locations on the farm [66]. The genera *Streptococcus*, *Acinetobacter*, *Pseudomonas*, *Staphylococcus*, *Sphingomonas*, *Enterobacter*, *Brachybacterium*, *Corynebacterium* detected in milk, were also found in various locations throughout the farm, including teat surfaces, milking parlors, hay, air, and dust. These findings showed that intramammary infection with these

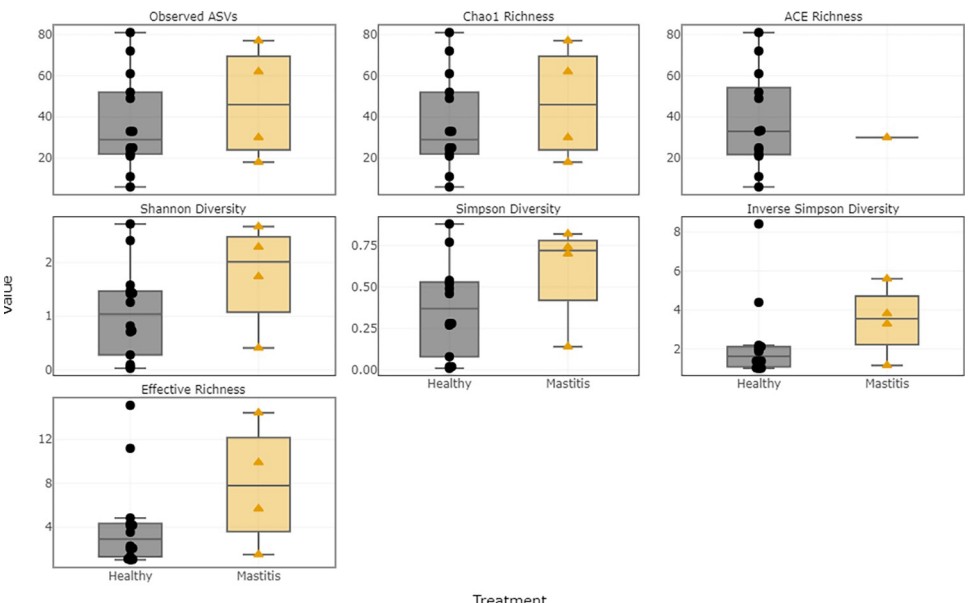

**Fig 10. Distribution of alpha diversity measures in relation to sample groups for the ITS.**

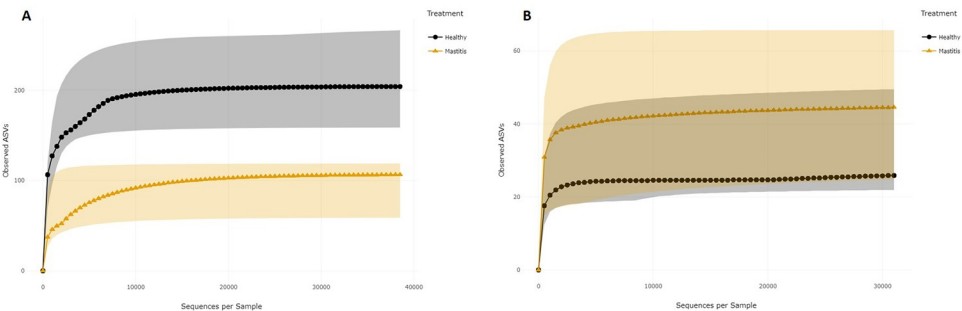

**Fig 11.** Rarefaction curves (median values) with interquartile ranges according to sample groups for the 16S rRNA (A) and for the ITS (B).

bacteria is mostly induced by crossing the teat canal from extramammary sites. However, *Lactococcus* and *Lactobacillus*, two technologically important bacteria, were found in cow milk but not in the dairy environment. Other factors such as improper hygiene practices and tick infestation may predispose camels to mastitis. Tick infestation is considered one of the primary causes of bacterial pathogenicity. The genera *Pseudomonas*, *Staphylococcus*, and *Jeotgalicoccus*

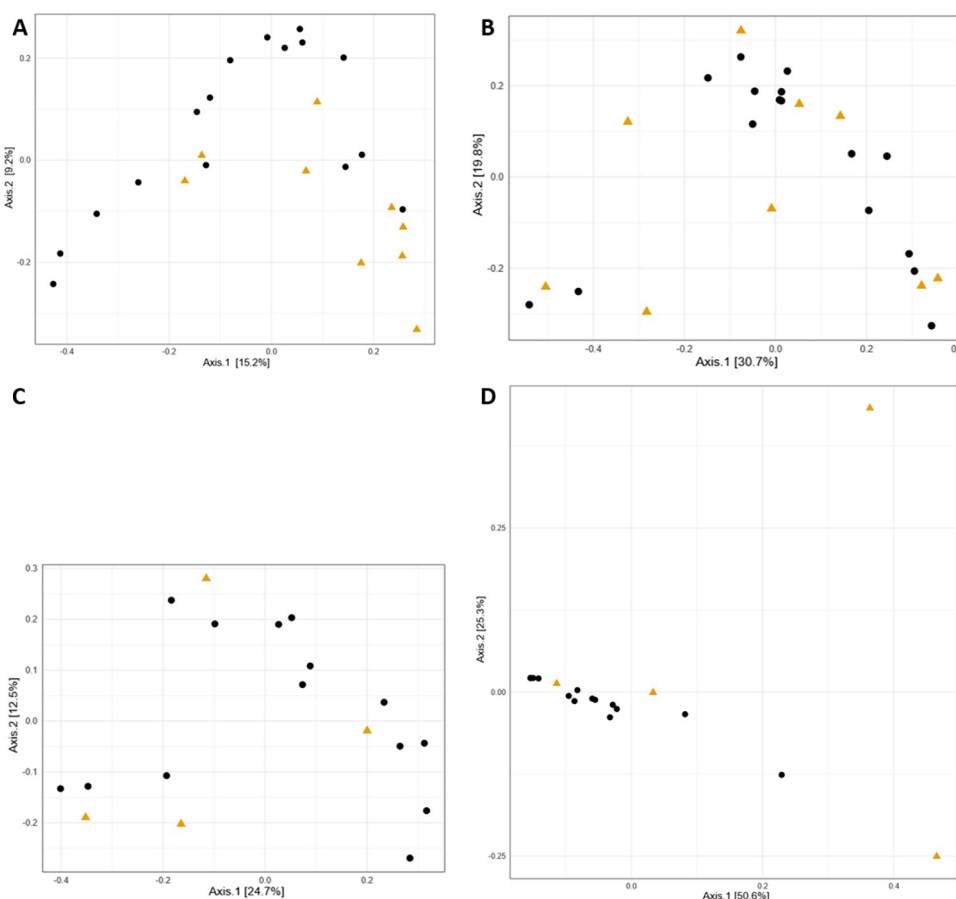

**Fig 12. PCoA visualizations on both unweighted and weighted UniFrac dissimilarities corresponding to healthy and mastitis samples.** A: unweighted UniFrac for the 16S rRNA, B: weighted UniFrac the 16S rRNA, C: unweighted UniFrac for the ITS, D: weighted UniFrac the ITS.

were among the wide diversity of microorganisms residing within ticks associated with camels in Saudi Arabia, highlighting the medical importance of ticks (*H. dromedarii*) as reservoir of pathogenic, opportunistic and symbiotic microorganisms [67].

In conclusion, this study provided an insight into the bacterial and fungal communities associated with camel mastitis. The results indicate that an imbalance or shift in the microbial population, as evidenced by an increase in the opportunistic bacteria and a considerable decrease in the commensal bacteria, is the primary cause of the development of camel mastitis. Additionally, higher bacterial diversity and richness and lower fungal diversity and richness are linked to camel mastitis. Future studies are needed to determine the fungal and bacterial pathogenesis in camel mastitis as well as the relationship between the microbiota and the factors predisposing camels to mastitis. Furthermore,

## Supporting information

**S1 Appendix.**
(PDF)

## Acknowledgments

The technical assistance of omics2view in sequencing and bioinformatics analysis is gratefully acknowledged.

## Author Contributions

**Conceptualization:** Rita Rahmeh.

**Formal analysis:** Rita Rahmeh, Abrar Akbar, Thnayan Alonaizi, Alfonso Esposito.

**Funding acquisition:** Rita Rahmeh.

**Investigation:** Abrar Akbar, Mohamed Kishk, Abdulaziz Al-Ateeqi, Anisha Shajan.

**Methodology:** Rita Rahmeh, Abrar Akbar, Husam Alomirah, Mohamed Kishk.

**Project administration:** Rita Rahmeh.

**Resources:** Mohamed Kishk, Abdulaziz Al-Ateeqi.

**Software:** Alfonso Esposito.

**Supervision:** Rita Rahmeh, Husam Alomirah.

**Validation:** Husam Alomirah.

**Visualization:** Rita Rahmeh.

**Writing – original draft:** Rita Rahmeh.

**Writing – review & editing:** Rita Rahmeh.

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
