## [Decision Letter · Decision Letter 0]

11 Oct 2022

PONE-D-22-24736Assessment of mastitis in camel using high-throughput sequencingPLOS ONE

Dear Dr. Rahmeh,

Thank you for submitting your manuscript to PLOS ONE. After careful consideration, we feel that it has merit but does not fully meet PLOS ONE’s publication criteria as it currently stands. Therefore, we invite you to submit a revised version of the manuscript that addresses the points raised during the review process.

In addition to the comments made by reviewers, I would like you to address the following questions and remarks. From the data presented, in the case of mastitis, it would have been interesting to compare the metagenomic data with more classical bacteriology methods. Can you infer the causative agent from the metagenomic analyses presented in Figure 2 ? One comment I have is that mastitis cases, although from different etiology as nicely shown in Figure 2, are pooled together in the results presented in Table 1, 2 and 3. Such a diversity in causes of mastitis is somehow hidden by the way data are presented.It might be worth specifying that the mastitis cases investigated in this report are clinical cases (based on symptoms and cell counts).I would also like you to explain the reason why no "no DNA control" was performed to characterize the ASVs that could originate from the processing of samples.L170: start sentence with capital "U".

We look forward to receiving your revised manuscript.

Kind regards,

Pierre Germon

Academic Editor

PLOS ONE

“This project was funded by Kuwait Foundation for the Advancement of Sciences (KFAS) under the project code PR18-12SL-16. The authors thank KFAS (Kuwait City, Kuwait), and Kuwait Institute for Scientific Research (Kuwait City, Kuwait) for their financial support. The technical assistance of omics2view in sequencing and bioinformatics analysis is gratefully acknowledged.”

“This project was funded by Kuwait Foundation for the Advancement of Sciences (KFAS) under the project code PR18-12SL-16”

Reviewers' comments:

Reviewer's Responses to Questions

**Comments to the Author**

1. Is the manuscript technically sound, and do the data support the conclusions?

Reviewer #1: Yes

Reviewer #2: Yes

2. Has the statistical analysis been performed appropriately and rigorously? 

Reviewer #1: Yes

Reviewer #2: Yes

3. Have the authors made all data underlying the findings in their manuscript fully available?

Reviewer #1: Yes

Reviewer #2: Yes

4. Is the manuscript presented in an intelligible fashion and written in standard English?

Reviewer #1: Yes

Reviewer #2: Yes

5. Review Comments to the Author

Reviewer #1: MAJOR COMMENTS

Camels are fundamental to food production in many arid regions in the world. Mastitis is a common and costly condition in camels, negatively affecting milk yield, quality and household income. Rahmeh et al have conducted a study, investigating the microbiome of milk from camels with healthy udders and milk from udders with clinical mastitis. This manuscript presents novel results regarding the microbial communities found in camel milk and specifically investigates the differences in microbiota in healthy and in mastitic milk.

The paper is scientifically sound, and methods are generally well described in sufficient detail. The sample size is quite modest which should be taken in consideration when interpreting the results. I do find it a bit of a concern that no milk samples were cultured, according to the manuscript, since this comparison would have provided very useful information about potential contamination at the time of sampling. This issue deserves to be addressed in the discussion.

The manuscript is overall well written but suffers from an abundance of minor linguistic errors which in some sections make the text difficult to interpret.

MINOR COMMENTS

Abstract

Lines 14-14: Mastitis could be caused by other pathogens as well, such as virus or algae.

Lines 18-19: Kindly specify what type of samples were taken.

Line 23: Maybe specify “in milk from inflamed udders” or something along those lines as it is not the milk that has mastitis.

Line 29-30: Same as above, this sentence needs to be clarified, maybe add “healthy milk and mastitic milk” to make it clearer?

Introduction

Line 39: Kindly remove the “s” from camelids.

Line 40: Change to “Asian and African regions” to be grammatically correct.

Line 40-41: These numbers are based on calculated estimates, especially in the major camel milk producing regions in East Africa, so I think this sentence should be phrased a bit more carefully.

Lines 41-44: What do the authors mean with “better than bovine milk”? Are these medicinal claims scientifically proven?

Line 44: Kindly change the tense of “increased” to be grammatically correct.

Line 53-54: I suggest the sentence is changed to “from one animal to another during the milking process” to be grammatically correct.

Line 55: Change to “The camel…” or “Camels…”

Line 56: Add a “the” before “symptoms”, change to “camels”

Lines 67-68: Remove “the” in front of "nomadic housing" and "improper hygienic measures".

Lines 70-72: Candida may be the leading cause of mycotic mastitis, but it is not the only fungus that could cause mastitis, please adjust this sentence.

Lines 75-76: Kindly change to “camels”. What do the authors mean by “little is known about pathogens involved in mastitis occurrence in camel” – that there is a small number of studies available? Because there is actually a rather large number of studies published investigating bacterial causes of both clinical and subclinical mastitis in dromedary camels. Or do the authors mean that there is limited knowledge available since most studies are based on conventional bacterial culture? Culturing is still the standard method for detection of intramammary infections in all dairy species. Kindly clarify to make to facilite for the readers.

Line 81: Please add “a” to “a high prevalence of fungi…”.

Line 87: Change to “have”.

Line 88: Please add “mastitis” after subclinical.

Line 89: Please add “a” in front of “next-generation…”.

Lines 95-97: These conclusive remarks should be saved for the Conclusions.

Materials and methods

Line 103: Kindly change to “camel owners’ “ as I assume there were more than one owner giving permission?

Sample collection

Line 105-106: I think “milking of individual dromedary camels…” would suffice as the sample selection is described more in detail further down.

Line 107: Kindly changed to “veterinarian”, and “localized” if that is what you mean.

Line 109: Was the SCC checked at the time of sampling? Kindly provide some more information about how the SCC procedure was carried out and why 200,000 cells/ml was chosen as a threshold for mastitis, as this is not mentioned anywhere in the introduction. Was SCC tested in all milk samples, also the ones taken from udders with clinical mastitis?

Line 111: Please add “was given” after “owner permission”. Please remove “the” in front of “disinfection”, change “were” to “was”.

Line 113-114: Was there any measures undertaken to keep the milk samples chilled?

DNA extraction

Line 117: A sentence should not begin with a number.

Statistical analysis

Line 156. Should be “P-values”, kindly check this throughout the manuscript.

Line 172: Please begin the sentence with a capital letter.

Results

Line 179-180: Kindly be consistent in how you write out numbers (like “twenty-five” or if you write them using digits like 16 and 9 in this sentence). Numbers smaller than twelve should be spelled out or else you should use digits, please check this throughout the manuscript. I think the origin of the samples could be more clearly stated. What is a “healthy sample”? Kindly check this throughout the manuscript (for ex. In the legends for figures 3&4).

Lines 182-183: Please clarify this sentence.

Lines 188-189: Please, clarify this sentence.

Camel milk microbiota in healthy and mastitis milk samples

Line 192: Kindly, add milk.

Line 200: What does “uncultured bacteria” refer to?

Line 222: “Condition” could be omitted. Please change “were” to “was” to improve the grammar.

Line 224: Substitute “in” with “with”.

Table 2: It would be helpful for interpretation if the unit was stated.

Line 234: Kindly replace percent with percentage.

Table 3&6. Both columns showing P-values (adjusted and unadjusted) contain identical values, maybe it would be sufficient with one column?

Line 258: Kindly substitute were for was.

Line 266: I think this sentence could be clarified, “In milk with mastitis…” or something like that as it doesn’t read right at present.

Lines 276 and 280: Should be percentage.

Line 281: Please, begin a sentence with a capital letter.

Figure 5&6: Why are only four samples shown in the mastitis category?

Line 286: Please clarify this sentence, the mastitis is still the inflammation of the mammary gland and the samples are not of the gland but of the milk.

Microbial Richness and Diversity in healthy and mastitis milk samples

Line 300: Change to lower case for all words in the heading to be consistent.

Lines 306 and 313-314: Please adjust these sentences so they are grammatically correct.

Line 310: Should be “were”.

Discussion

Line 354: Kindly, change had to have.

Line 356: Should there not be a “milk” at the end of this sentence?

Line 358: Change “is” to “are”.

Line 369: Should be “camels”.

Line 373: replace the “,” with “and” (Staphylococcus and Streptococcus…”)

Line 374: Both of the studies referred to are from Ethiopia, kindly remove Kenya or add a reference from Kenya.

Line 379: Should be “humans”.

Line 385: Add “in” before “bovine milk” and “buffalo milk”.

Line 389: Add “a” before “previous study”.

Lines 403-409: Maybe clarify this section so that the readers understand that these results are not originating from the camel dairy industry.

Line 410: Please, begin the sentence with a capital letter.

Line 413: Goat mastitis is the same as caprine mastitis.

Line 413: Mastitis is not produced?

Line 418-419: Kindly adjust this sentence to make it grammatically correct.

Line 437: Omit “the” in front of “improper milking…”.

Lines 438-439: The significance of tick infestation as the “major cause” of mastitis needs to be supported with adequate references.

Reviewer #2: Regarding manuscript entitled “Assessment of mastitis in camel using high-throughput sequencing”, the authors aimed to report the bacterial and fungal community involved in camel mastitis using Illumina amplicon sequencing.

The study is interesting and can be accepted after minor revision. Pleases add some details regarding camel age, parity and management.

6. PLOS authors have the option to publish the peer review history of their article (what does this mean?). If published, this will include your full peer review and any attached files.

Reviewer #1: No

Reviewer #2: **Yes: **Ayman A Swelum

---

## [Author Response · Author response to Decision Letter 0]

1 Nov 2022

Response to Academic Editor

• From the data presented, in the case of mastitis, it would have been interesting to compare the metagenomic data with more classical bacteriology methods. Can you infer the causative agent from the metagenomic analyses presented in Figure 2? One comment I have is that mastitis cases, although from different etiology as nicely shown in Figure 2, are pooled together in the results presented in Table 1, 2 and 3. Such a diversity in causes of mastitis is somehow hidden by the way data are presented.

AU: We thank the editor for his careful reading of the manuscript and for his constructive comments. Complementing culture-independent and culture-based approaches provide a complete understanding of the disease etiology. Culture-dependent approaches only target cultivable bacteria and in some cases, have insufficient discriminatory power. Isolates will strongly depend on the media used, sample storage, and growth conditions. Many microorganisms are averse to isolation using common culturing methods, thus potentially leading to a significant underestimation of the microbial communities. Considering this major bias, high-throughput sequencing approaches detect DNA from all the bacteria present in a sample, alive or not, allowing a much more in-depth and accurate estimation of microbial diversity. It is misleading to infer, from such kind of data (culture-based), if the causative agent is a single species just because its abundance is higher or because it has been found in previous analyses. There is increasing evidence that dysbiotic states are not only connected to a single species, rather to changes in the composition and function of the whole microbiome (i.e. conceptualized in the term “pathobiome” as defined in Bass et al 2019 (10.1016/j.tree.2019.07.012)). Our study points toward this approach, consider the microbiome as a whole and determine the difference between healthy and mastitis microbiome.

Figure 2 gives an insight into the mastitis-causing pathogens and demonstrates a diversity in camel mastitis etiology. As suggested, Tables S4 and S5 summarizing the taxon frequencies of the most dominant bacterial population in all the healthy and mastitis camel milk samples at the phylum and genus levels are now added to the supporting information.

We totally agree with the editor, further study is required to fully understand camel mastitis, which is now addressed in the discussion lines 404-408: “It is noteworthy that this study demonstrates a diversity in camel mastitis etiology. The increase in the abundance of Staphylococcus and Streptococcus is mainly associated to two samples CM-69M and CM-70M originating from camels suffering from severe clinical mastitis. To fully understand the mastitis-causing pathogens in camels, additional investigations are required, including a large number of mastitic milk samples along with bacterial culturing”. 

• It might be worth specifying that the mastitis cases investigated in this report are clinical cases (based on symptoms and cell counts).

AU: We thank the editor for this comment. Indeed, it is clinical mastitis, we added this information in the abstract lines 18 and 32.

• I would also like you to explain the reason why no "no DNA control" was performed to characterize the ASVs that could originate from the processing of samples.

AU: All healthy and mastitis samples were processed in the same way at the same time during the entire workflow (DNA extraction, library preparation, sequencing, and bioinformatics analysis). Any background noise, if existing, may thus be present in both conditions (healthy and mastitis) which will not affect the results generated from their comparison. 

L170: start sentence with capital "U".

AU: Corrected

AU: The manuscript style requirements was verified

“This project was funded by Kuwait Foundation for the Advancement of Sciences (KFAS) under the project code PR18-12SL-16. The authors thank KFAS (Kuwait City, Kuwait), and Kuwait Institute for Scientific Research (Kuwait City, Kuwait) for their financial support. The technical assistance of omics2view in sequencing and bioinformatics analysis is gratefully acknowledged.”

“This project was funded by Kuwait Foundation for the Advancement of Sciences (KFAS) under the project code PR18-12SL-16”

AU: The funding-related text is now removed from the manuscript. The current funding statement is correct.

AU: The reference list was reviewed.

Response to Reviewers

Reviewer #1: MAJOR COMMENTS

Camels are fundamental to food production in many arid regions in the world. Mastitis is a common and costly condition in camels, negatively affecting milk yield, quality and household income. Rahmeh et al have conducted a study, investigating the microbiome of milk from camels with healthy udders and milk from udders with clinical mastitis. This manuscript presents novel results regarding the microbial communities found in camel milk and specifically investigates the differences in microbiota in healthy and in mastitic milk.

The paper is scientifically sound, and methods are generally well described in sufficient detail. 

• The sample size is quite modest which should be taken in consideration when interpreting the results. I do find it a bit of a concern that no milk samples were cultured, according to the manuscript, since this comparison would have provided very useful information about potential contamination at the time of sampling. This issue deserves to be addressed in the discussion.

AU: We thank the reviewer for the careful reading of the manuscript and for helpful and constructive suggestions that greatly improved the manuscript. We agree with the reviewer, the sample size was modest due to 1) the limited number of camels suffering from clinical mastitis and showing obvious signs of mastitis on their udder and milk, 2) the quantity of extracted DNA was very low for some samples, 3) the library preparation was not successful for some samples. 

Compared to the culturing methods, culture-independent approaches are more adequate to identify the mastitis-causing pathogens. There is increasing evidence that dysbiotic states are not only connected to a single species, rather to changes in the composition and function of the whole microbiome (i.e. conceptualized in the term “pathobiome” as defined in Bass et al 2019 (10.1016/j.tree.2019.07.012)). Our study points toward this approach, consider the microbiome as a whole and determine the difference between healthy and mastitis microbiome. However, the reviewer is totally right; this study gave insights about the possible causative agents of mastitis in camel and further complementary study is required. This issue is now addressed in the discussion lines 404-408: “It is noteworthy that this study demonstrates a diversity in camel mastitis etiology. The increase in the abundance of Staphylococcus and Streptococcus is mainly associated to two samples CM-69M and CM-70M originating from camels suffering from severe clinical mastitis. To fully understand the mastitis-causing pathogens in camels, additional investigations are required, including a large number of mastitic milk samples along with bacterial culturing”. 

Furthermore, in the discussion, a paragraph about the bacteria that were suggested as the main cause of camel mastitis identified by culture-based methods in other studies is mentioned on page 16 (lines 385-389). 

• The manuscript is overall well written but suffers from an abundance of minor linguistic errors which in some sections make the text difficult to interpret.

AU: The manuscript has been edited and revised taking into consideration all the linguistic errors

MINOR COMMENTS

Abstract

Lines 14-14: Mastitis could be caused by other pathogens as well, such as virus or algae.

AU: The sentence was reformulated to mastitis in general because we are not sure if there is available information about the other pathogens such as virus and algae as causative agents of camel mastitis. 

Lines 18-19: Kindly specify what type of samples were taken.

AU: Added.

Line 23: Maybe specify “in milk from inflamed udders” or something along those lines as it is not the milk that has mastitis.

AU: Added.

Line 29-30: Same as above, this sentence needs to be clarified, maybe add “healthy milk and mastitic milk” to make it clearer?

AU: As suggested, the sentence is now clarified

Introduction

Line 39: Kindly remove the “s” from camelids.

AU: Removed.

Line 40: Change to “Asian and African regions” to be grammatically correct.

AU: Changed.

Line 40-41: These numbers are based on calculated estimates, especially in the major camel milk producing regions in East Africa, so I think this sentence should be phrased a bit more carefully.

AU: As suggested, the sentence was rephrased.

Lines 41-44: What do the authors mean with “better than bovine milk”? Are these medicinal claims scientifically proven?

AU: Yes, compared to bovine milk, camel milk’s digestibility is higher. It has a higher nutritional value, and contains a higher amount of proteins that have a positive effect on the immune system. Camel milk possesses little or no allergy effects, and lactose-intolerant people have no difficulties in metabolizing its lactose. Reference: Khalesi, Mohammadreza, et al. "Biomolecular content of camel milk: A traditional superfood towards future healthcare industry." Trends in Food Science & Technology 62 (2017): 49-58.

Line 44: Kindly change the tense of “increased” to be grammatically correct.

AU: Changed.

Line 53-54: I suggest the sentence is changed to “from one animal to another during the milking process” to be grammatically correct.

AU: Changed.

Line 55: Change to “The camel…” or “Camels…”

AU: Changed.

Line 56: Add a “the” before “symptoms”, change to “camels”

AU: Changed.

Lines 67-68: Remove “the” in front of "nomadic housing" and "improper hygienic measures".

AU: Removed.

Lines 70-72: Candida may be the leading cause of mycotic mastitis, but it is not the only fungus that could cause mastitis, please adjust this sentence.

AU: we agree with the reviewer; the sentence is now adjusted.

Lines 75-76: Kindly change to “camels”. What do the authors mean by “little is known about pathogens involved in mastitis occurrence in camel” – that there is a small number of studies available? Because there is actually a rather large number of studies published investigating bacterial causes of both clinical and subclinical mastitis in dromedary camels. Or do the authors mean that there is limited knowledge available since most studies are based on conventional bacterial culture? Culturing is still the standard method for detection of intramammary infections in all dairy species. Kindly clarify to make to facilite for the readers.

AU: We thank the reviewer for this comment. Yes, we mean that there is limited knowledge available since most studies are based on conventional bacterial culture. It is now clarified.

Line 81: Please add “a” to “a high prevalence of fungi…”.

AU: Added.

Line 87: Change to “have”.

AU: Changed.

Line 88: Please add “mastitis” after subclinical.

AU: Added.

Line 89: Please add “a” in front of “next-generation…”.

AU: Added.

Lines 95-97: These conclusive remarks should be saved for the Conclusions.

AU: we agree with the reviewer; these conclusive remarks are now deleted.

Materials and methods

Line 103: Kindly change to “camel owners’ “ as I assume there were more than one owner giving permission?

AU: Changed.

Sample collection

Line 105-106: I think “milking of individual dromedary camels…” would suffice as the sample selection is described more in detail further down.

AU: Changed.

Line 107: Kindly changed to “veterinarian”, and “localized” if that is what you mean.

AU: Changed.

Line 109: Was the SCC checked at the time of sampling? Kindly provide some more information about how the SCC procedure was carried out and why 200,000 cells/ml was chosen as a threshold for mastitis, as this is not mentioned anywhere in the introduction. Was SCC tested in all milk samples, also the ones taken from udders with clinical mastitis?

AU: We thank the reviewer for this comment. A paragraph was added in the methodology including the suggested information: The SCC count for all milk samples was performed immediately upon arrival at the laboratory using a direct cell counter (DCC, DeLaval International AB). The optimum SCC threshold for healthy milk is estimated to be around 200,000 cells/mL (Scheperz, 1997, Karzis et al., 2017). The milk samples with SCC less than 200,000 cells/ml were considered healthy milk samples.

National Mastitis Council Guidelines: Clinical mastitis is, by definition, abnormal milk, and no reference to SCC is required. However, clinical mammary quarters will almost always have SCC greater than 200,000 cells/ml (https://www.nmconline.org/wp content/uploads/2016/09/Guidelines-on-Normal-and-Abnormal-Raw-Milk.pdf).

Line 111: Please add “was given” after “owner permission”. Please remove “the” in front of “disinfection”, change “were” to “was”.

AU: Added.

Line 113-114: Was there any measures undertaken to keep the milk samples chilled?

AU: Yes, the samples were transported in an icebox.

DNA extraction

Line 117: A sentence should not begin with a number.

AU: Edited.

Statistical analysis

Line 156. Should be “P-values”, kindly check this throughout the manuscript.

AU: Corrected.

Line 172: Please begin the sentence with a capital letter.

AU: Edited.

Results

Line 179-180: Kindly be consistent in how you write out numbers (like “twenty-five” or if you write them using digits like 16 and 9 in this sentence). Numbers smaller than twelve should be spelled out or else you should use digits, please check this throughout the manuscript. I think the origin of the samples could be more clearly stated. What is a “healthy sample”? Kindly check this throughout the manuscript (for ex. In the legends for figures 3&4).

AU: The numbers are now checked in all the manuscript. As suggested, the origin of samples and the criteria for healthy samples is now added in the methods section (lines 121-123) and in the results section (188-189). 

Lines 182-183: Please clarify this sentence.

AU: Clarified

Lines 188-189: Please, clarify this sentence.

AU: Clarified

Camel milk microbiota in healthy and mastitis milk samples

Line 192: Kindly, add milk.

AU: Added

Line 200: What does “uncultured bacteria” refer to?

AU: Definition is now added. 

Line 222: “Condition” could be omitted. Please change “were” to “was” to improve the grammar.

AU: Changed

Line 224: Substitute “in” with “with”.

AU: Substituted

Table 2: It would be helpful for interpretation if the unit was stated.

AU: The unit (%) is now stated

Line 234: Kindly replace percent with percentage.

AU: Replaced

Table 3&6. Both columns showing P-values (adjusted and unadjusted) contain identical values, maybe it would be sufficient with one column?

AU: We agree with the reviewer; one column is now kept.

Line 258: Kindly substitute were for was.

AU: Substituted

Line 266: I think this sentence could be clarified, “In milk with mastitis…” or something like that as it doesn’t read right at present.

AU: As suggested, this sentence is now edited

Lines 276 and 280: Should be percentage.

AU: Changed

Line 281: Please, begin a sentence with a capital letter.

AU: Changed

Figure 5&6: Why are only four samples shown in the mastitis category?

AU: For the ITS analysis, the library preparation for sequencing was successful for four samples. 

Line 286: Please clarify this sentence, the mastitis is still the inflammation of the mammary gland and the samples are not of the gland but of the milk.

AU: Clarified

Microbial Richness and Diversity in healthy and mastitis milk samples

Line 300: Change to lower case for all words in the heading to be consistent.

AU: Changed

Lines 306 and 313-314: Please adjust these sentences so they are grammatically correct.

AU: Adjusted

Line 310: Should be “were”.

AU: Changed

Discussion

Line 354: Kindly, change had to have.

AU: Changed

Line 356: Should there not be a “milk” at the end of this sentence?

AU: Added

Line 358: Change “is” to “are”.

AU: Changed

Line 369: Should be “camels”.

AU: s is added

Line 373: replace the “,” with “and” (Staphylococcus and Streptococcus…”)

AU: Replaced

Line 374: Both of the studies referred to are from Ethiopia, kindly remove Kenya or add a reference from Kenya.

AU: We agree with the reviewer, Kenya was removed

Line 379: Should be “humans”.

AU: “s” is added

Line 385: Add “in” before “bovine milk” and “buffalo milk”.

AU: Added

Line 389: Add “a” before “previous study”.

AU: Added

Lines 403-409: Maybe clarify this section so that the readers understand that these results are not originating from the camel dairy industry.

AU: As suggested, this section is now clarified

Line 410: Please, begin the sentence with a capital letter.

AU: Done

Line 413: Goat mastitis is the same as caprine mastitis.

AU: This sentence was reformulated to make it clearer

Line 413: Mastitis is not produced?

AU: This sentence was reformulated to make it clearer

Line 418-419: Kindly adjust this sentence to make it grammatically correct.

AU: Edited

Line 437: Omit “the” in front of “improper milking…”.

AU: Deleted

Lines 438-439: The significance of tick infestation as the “major cause” of mastitis needs to be supported with adequate references.

AU: Tick infestation is considered one of the primary causes of bacterial pathogenicity (not mastitis in particular), the reference is the same as the one for the following sentence (reference 67). Since the genera Pseudomonas, Staphylococcus, and Jeotgalicoccus were among the wide diversity of microorganisms residing within ticks associated with camels in Saudi Arabia, tick infestation may predispose camels to mastitis.

Reviewer #2: Regarding manuscript entitled “Assessment of mastitis in camel using high-throughput sequencing”, the authors aimed to report the bacterial and fungal community involved in camel mastitis using Illumina amplicon sequencing.

The study is interesting and can be accepted after minor revision. Pleases add some details regarding camel age, parity and management.

AU: We thank the reviewer for his time and interest in our study. A table is now added to the supporting information including the camel age, parity and management (S1 Table).

---

## [Editor Report · Decision Letter 1]

17 Nov 2022

Assessment of mastitis in camel using high-throughput sequencing

PONE-D-22-24736R1

Dear Dr. Rahmeh,

We’re pleased to inform you that your manuscript has been judged scientifically suitable for publication and will be formally accepted for publication once it meets all outstanding technical requirements.

Kind regards,

Pierre Germon

Academic Editor

PLOS ONE
---

## [Editor Report · Acceptance letter]

28 Nov 2022

PONE-D-22-24736R1 

Assessment of mastitis in camel using high-throughput sequencing 

Dear Dr. Rahmeh:

I'm pleased to inform you that your manuscript has been deemed suitable for publication in PLOS ONE. Congratulations! Your manuscript is now with our production department. 

Kind regards, 

on behalf of

Dr. Pierre Germon 

Academic Editor

PLOS ONE